# Lipoprotein(a) levels in a sample of 115,197 subjects from the largest Brazilian private laboratory

Maria Helane Costa Gurgel Castelo [1], Isac de Castro[1], Priscila Raupp-da-Rosa [2*], Patrícia Cristina Grenzi[3], Eduardo Gomes Lima[4], Andreza Almeida Senerchia[4], Érica Ferreira[5], Flavia Paiva Proença Lobo Lopes[6]

1 Clinical Pathology Department, DASA, São Paulo, SP, Brazil, 2 Evidence Generation, Novartis Biociências S.A, São Paulo, SP, Brazil, 3 External Scientific Engagement, Alexion Pharmaceuticals, São Paulo, SP, Brazil, 4 Clinical Research, DASA, São Paulo, SP, Brazil, 5 Study & Site Operations (SSO) Study Start-Up, Novartis Biociências S.A., São Paulo, SP, Brazil, 6 Research, Education and Innovation Department, DASA, São Paulo, SP, Brazil

* Priscila.raupp@novartis.com

## Abstract

Lipoprotein(a) (Lp(a)) is an independent risk factor for atherosclerotic disease and is increasingly being incorporated into clinical algorithms of cardiovascular risk prediction. However, the epidemiology of Lp(a) in Brazil remains unknown. The objective of this study was to describe the distribution of Lp(a) levels and its association with laboratory parameters and clinical characteristics in a population of subjects submitted to blood tests in a private laboratory. Methods involved assessing Lp(a) levels from 115,197 subjects in a nationwide database from one Brazilian private laboratory, with Lp(a) measured using a nephelometric assay and expressed in mg/dL. Results showed that among the 115,197 subjects, the median age was 44 years. Women composed 61% of the sample and displayed higher Lp(a) levels in comparison to men (13.90 vs 11.58 mg/dL, p < 0.001). The distribution of Lp(a) levels was skewed rightward, as 70% of the individuals showed levels < 30 mg/dL, and 18% had levels higher than 50 mg/dL. The presence of criteria for diabetes, metabolic syndrome, and levels of HDL-cholesterol and triglycerides showed no correlation with Lp(a) levels. Additionally, there was no significant association between Lp(a) levels in individuals with and without criteria for diabetes and metabolic syndrome. In conclusion, this study represents the largest descriptive analysis of Lp (a) in Brazil, encompassing individuals with a wide age range and geographical distribution. This data contributes to the generation of high-quality evidence needed for improving cardiovascular risk assessment in clinical care settings.

**Data availability statement:** All relevant data are within the manuscript and its Supporting Information files.

**Funding:** Funding Statement: Novartis Biociências SA provided financial support for the conduct of the research, preparation of the article, study design, critical analysis, interpretation of data, writing of the report, and the decision to submit the article for publication. There was no additional external funding received for this study.

**Competing interests:** The authors have disclosed the following potential conflicts of interest: PR is an employee and stockholder of Novartis Biociências SA. PR also holds a leadership role as an Emergent Leader in the World Heart Federation. EGL reports that Novartis made payments to DASA, their current employer, in relation to this manuscript. EGL also received honoraria as a speaker for Novartis in educational events and has served on an advisory board for Bayer. FPPLL received direct payments from Bayer for attending meetings or travel. EF is a salaried employee of Novartis Biociências SA. IC received consulting fees through their institution, ConstatBio, related to DASA. PCG was a salaried employee of Novartis Biociências SA during the preparation of this manuscript. MHCGC and AAS declared no competing interests. These commercial affiliations do not alter our adherence to PLOS ONE's policies on data and material sharing.

## Introduction

Lipoprotein(a) (Lp(a)) is a type of low-density lipoprotein (LDL) particle synthesized by the liver, consisting of both apolipoprotein(a) and apolipoprotein B proteins [1]. Epidemiological findings have established a clear link between Lp(a) and several cardiovascular diseases, encompassing conditions like myocardial infarction, stroke, and aortic valve stenosis [2]. Furthermore, genetic evidence, employing the Mendelian randomization approach, has offered compelling support for the causality of these associations [2].

As more evidence emerges, Lp(a) is gaining greater recognition as a significant factor in cardiovascular risk assessment, leading to its growing recommendation for measurement in recent guidelines [3,4]. The standard distributions of Lp(a) in the general Caucasian population have been established based on findings from the Copenhagen City Heart Study [5]. In this study, a sample of 20,000 individuals was initially taken, and a subgroup analysis of 3,000 men and 3,000 women revealed that approximately 20% of participants had plasma Lp(a) levels higher than 50 mg/dL, representing individuals above the 80th percentile [6].

The Emerging Risk Factors Collaboration [7] has shown that serum Lp(a) levels exhibit remarkable consistency within individuals over the course of several years, with no significant correlations observed between its levels and lifestyle modifications or other risk factors. While its structure exhibits significant heterogeneity, the stability of Lp(a) levels through an individual's lifespan is primarily determined by genetics, for 80–90% of Lp(a) variability is genetically determined [8].

Therefore, it is recommended that Lp(a) levels need only to be measured once in a lifetime [3,4] unless a secondary cause of alteration in Lp(a) levels, such as the influence of physical activity, hormone replacement therapy, kidney diseases, or liver diseases, is suspected or if specific treatment is initiated to lower its levels [9].

Current guidelines suggest that measurement of Lp(a) should be considered in patients with early atherosclerotic cardiovascular disease (ASCVD), a family history of elevated Lp(a) or premature ASCVD, Familial Hypercholesterolemia, recurrent cardiovascular events despite the use of statins and precocious calcific aortic stenosis [3,4].

The epidemiological landscape of Lp(a) in Brazil remains largely unexplored and undisclosed at present. A comprehensive understanding of Lp(a) prevalence, distribution, and associated factors within the Brazilian population is currently lacking, highlighting the need for dedicated research efforts. To gain an understanding of Lp(a) levels and their associations with lipid fractions and clinical conditions, offering valuable insights into the population-level distribution of Lp(a), we undertook a retrospective observational analysis. This study utilized data from a nationwide database sourced from a Brazilian private laboratory, allowing us to elucidate the demographic and laboratory parameters of patients who underwent testing for Lp(a) levels.

## Methods

### Study population and design

This is a descriptive, retrospective study, using real-world data from a Brazilian private laboratory, called Diagnósticos da América S.A. (DASA), database.

For the present analysis, we examined registered data from 5,016,790 patients assessed from centers distributed all over the Brazilian territory. Biochemical measurements were performed between January 2016 and November 2019. Due to the retrospective design, a patient could contribute one or multiple samples. In this instance, we deemed appropriate to include the highest registered level of Lp(a). Any missing demographic or laboratory data was considered an exclusion criterion.

## Clinical outcomes

All our clinical definitions were exclusively derived from laboratorial data. According to the Brazilian Society of Diabetes criteria [10] – prediabetes was defined as follows: fasting plasma glucose ≥ 100 mg/dL and < 200 mg/dL (1 exam) or HbA1cc ≥ 5.7% and < 6.5% (1 exam); and Diabetes Mellitus (DM) was considered when: random plasma glucose ≥ 200 mg/dL (1 exam) or fasting plasma glucose ≥ 126 mg/dL (2 non-consecutive exams over 12 months) or HbA1c ≥ 6.5%(1 exam).

Based on US National Cholesterol Education Program (NCEP) definitions [11], the following metabolic syndrome laboratorial criteria were considered: triglycerides ≥ 150 mg/dL; fasting plasma glucose ≥ 110 mg/dL; HDL-cholesterol < 40 mg/dL (men), HDL-cholesterol < 50 mg/dL (women). Furthermore, in line with NCEP definitions, LDL-c levels > 130 mg/dL were considered elevated [11]. Patients with Familiar Hypercholesterolemia were not excluded from the analysis.

Our analysis of Lp(a) levels focused on four main thresholds of interest: 30 mg/dL, 50 mg/dL, 70 mg/dL and 90 mg/dL. The 30 mg/dL threshold is classically considered the threshold of normality, from which cardiovascular risk starts to rise progressively [3]. The 50 mg/dL threshold has been considered a risk enhancer, favoring the intensification of lifestyle interventions and initiation of statin therapy [3,4]. The 70 mg/dL and 90 mg/dL thresholds are currently under investigation in the L(a)HORIZON cardiovascular outcomes trial [12], that bears the potential for introducing novel specific Lp(a)-lowering interventions into clinical practice, with results expected in 2025.

## Database

The database consists of, but is not limited to, demographic factor (age, gender, geographic location), basic and comprehensive metabolic panels (glucose, HbA1c), lipid panels (LDL-cholesterol, HDL-cholesterol, Lp(a), triglycerides). These results are also linked to an individual electronic medical record, which can be manipulated (i.e., filtered, extracted, and analyzed) to monitor laboratory parameters and, if relevant, to link the results to specific diseases, conditions, and ICD codes for patients admitted to hospitals maintained by DASA.

## Laboratorial measurements

The following laboratory blood tests were analyzed in study subjects: fasting blood glucose, HbA1c, total plasma cholesterol, HDL cholesterol, LDL cholesterol, triglycerides, Lp (a). Lp(a) was measured in mg/dL by nephelometric assay in Cobas 8000, Imuno BNII Cluster 1, Cluster 2, and Main Cluster devices. Levels below 30 mg/dL were considered normal. Fasting plasma glucose was evaluated by the hexokinase method, using NTO Alpha Bioquimica Advia Cluster, reference interval: 70–99 mg/dL. HbA1c was measured by Advia Cluster, Cobas, Premier Hb9210, Tosoh Cluster, Cluster GHLV, Hedmato D-100 and Cobas C513. Reference ranges from < 5.7% (normal); 5.7% to 6.5% (elevated risk of DM); and > 6.5% (DM).

Triglyceride's were measured using the In Advia 7 Standalone, Advia cluster, Cobas C502, Cobas 8000, Modular P and Olympus equipment. TG was measured using an enzymatic colorimetric method with cholesterol esterase oxidase. Reference ranges for TG were < 150 mg/dL (if fasting), or < 175 mg/dL (no fasting). Total plasma cholesterol was measured by enzymatic colorimetric method with glycerol phosphate oxidase. Advia Cluster, Cobas, Modular P, Olympus, Cobas C503 and Cobas 8000 equipment was used. Levels < 190 mg/dL were considered desirable.

LDL-C assay was performed In Advia 7 Standalone, Advia cluster, Cobas C502, Cobas 8000, Modular P and Olympus equipment. From January 2016 until March 2019, LDL-cholesterol was calculated using Friedewald formula. From April 2019 to November 2019, Martin-Hopkins Formula was used. Reference ranges for LDL-cholesterol were < 110 mg/dL

(with/without fasting). HDL-C was measured through homogeneous assay. Samples were processed in Cluster, Cobas, Modular P, Olympus, and Cobas C502 equipment. Reference ranges: > 40 mg/dL.

## Statistical analysis

Continuous data, as Lp(a) (mg/dL), LDL-C (mg/dL), HDL-C (mg/dL), Triglycerides (mg/dL); and semi-continuous, such as age (complete years), were compared with the Gauss curve and determined as non-parametric through the K-S Distance test (Kolmogorov-Smirnov) and therefore were represented in median and percentiles (25–75).

When comparing two independent groups (gender comparison), the Mann-Whitney test with Bonferroni correction was used. When comparing three or more groups, the Kruskal-Wallis test with Bonferroni correction and the modified Kolmogorov-Smirnov contrast post-test were used.

Levels of LDL-C (mg/dL), HDL-C (mg/dL), triglycerides (mg/dL) and age (complete years) were categorized and represented by absolute (n) and relative (%) frequency. The contingency matrices were analyzed by Pearson's chi-square test.

We chose to correct the continuous and semi-continuous data described in median and percentiles using the "bootstrap resample" technique with 1,000 resampling's. When performing the comparison tests between groups, the Monte Carlo correction was performed with a thousand resampling. Resampling values were represented by upper and lower 95% confidence intervals (CI).

For the analysis of distribution of Lp(a), non-linear regression was performed adjusted to a third-order polynomial model. This model was chosen because it presents the highest linearity value ($r^2$) with low residual value. The nonlinear regression model was performed for the values of percentiles 90 and 95% as it is an estimator of the limit of normality of the population in terms of Lp(a).

For the entire study, an alpha risk of less than or equal to 5% of committing a type I or 1st kind error and a Beta risk of less than or equal to 20% of committing a type II or 2nd kind error was considered.

## Ethical considerations

During clinical visits, patients voluntarily sign a privacy agreement, granting consent for generating analyses and studies that contribute to the improvement of DASA's activities in compliance with its Privacy Policy and the General Data Protection Law in Brazil. This protocol was also submitted to the appreciation of the local Research Ethics Committee, under the reference number 45538021.1.0000.5455. Due to the large number of patients and the retrospective nature of the study, a waiver of informed consent was obtained. The participants identity and personal data was not shared, and just aggregated data has been reported. This study follows the principles of Good Clinical Practices and the ethical principles referred in the Declaration of Helsinki. All data was anonymized and encrypted.

## Results

From a total of 5,016,790 patients and approximately 8 million exams registered during the study period, complete data was available for a subset of 115,192 individuals totaling 143,397 Lp (a) measurements.

Subjects were predominantly young and middle-aged adults, since 45% of them were aged from 30 to 50 years. Women constituted most of the sample (n = 70,550 individuals, 61.2% of the total) and displayed higher Lp(a) levels in comparison to men (13.90 vs 11.58 mg/dL, p varied from < 000.1 to 0.003, Table 1).

A geographical analysis of patient locations across the country revealed that these individuals were primarily concentrated in urban centers such as São Paulo, Rio de Janeiro, Brasília, Curitiba, Recife, and Fortaleza, respectively.

Lp(a) distribution according to gender and age groups is displayed in Table 1. Our analysis indicated a median Lp(a) level of 12.95 mg/dL (interquartile interval 4.54–36.6 mg/dL; p varied from < 0.0001 to 0.003, Table 1). In the sample, 30% of the participants displayed Lp(a) levels surpassing 30 mg/dL, with a cumulative count of 34,082 individuals. Based on our dataset, 18% of the individuals (20,859 subjects) had Lp(a) levels above 50 mg/dL, which is considered a high risk for

**Table 1. Distribution of Lp(a), LDL-cholesterol, HDL-cholesterol and triglycerides levels in different genders and age groups.**

| | | | Lp(a) | | | LDL-C | | | TG | | | HDL-C | | |
|---|---|---|---|---|---|---|---|---|---|---|---|---|---|---|
| **Total** | | | Median | 25% | 75% | Median | 25% | 75% | Median | 25% | 75% | Median | 25% | 75% |
| **n** | 115.197 | 100% | **12.95** | 4.54 | 36.60 | **107.20** | 84.72 | 132.00 | **98.40** | 71.00 | 141.00 | **53.00** | 44.00 | 64.00 |
| **Gender** | | | | | | | | | | | | | | |
| **Female** | 70550 | 61.2% | **13.90** | 4.84 | 38.19 | **106.20** | 84.84 | 130.54 | **91.00** | 66.00 | 128.00 | **58.20** | 49.00 | 69.00 |
| **Male** | 44647 | 38.8% | **11.58** | 4.10 | 33.80 | **108.96** | 84.46 | 134.60 | **112.00** | 80.00 | 160.00 | **46.00** | 39.00 | 54.00 |
| **Mann-Whitney Sig.** | | | < 0.0001 | | | < 0.0001 | | | < 0.0001 | | | < 0.0001 | | |
| **95%CI Inf. Lim.** | | | < 0.0001 | | | < 0.0001 | | | < 0.0001 | | | < 0.0001 | | |
| **95%CI Sup. Lim.** | | | 0.0030 | | | 0.0021 | | | 0.0025 | | | 0.0025 | | |
| **Age Groups** | | | | | | | | | | | | | | |
| **1 - 10** | 602 | 0.52% | **14.65** | 4.08 | 42.50 | **93.54** | 79.36 | 111.80 | **71.00** | 54.80 | 98.60 | **53.00** | 45.00 | 62.10 |
| **11 - 20** | 3791 | 3.29% | **12.92** | 4.50 | 37.30 | **88.60** | 72.70 | 108.00 | **76.00** | 58.00 | 105.00 | **51.00** | 43.30 | 61.00 |
| **21 - 30** | 14068 | 12.22% | **11.50** | 4.23 | 33.02 | **100.20** | 81.80 | 122.18 | **82.00** | 60.00 | 114.00 | **55.00** | 45.00 | 66.20 |
| **31 - 40** | 29736 | 25.83% | **11.50** | 4.17 | 32.88 | **108.00** | 87.68 | 131.40 | **88.00** | 64.00 | 127.00 | **53.00** | 43.90 | 64.30 |
| **41 - 50** | 23570 | 20.47% | **12.50** | 4.42 | 35.13 | **114.40** | 92.37 | 137.65 | **99.20** | 72.00 | 145.00 | **52.90** | 43.40 | 64.00 |
| **51 - 60** | 19947 | 17.33% | **14.42** | 4.88 | 40.21 | **115.82** | 91.20 | 141.20 | **112.40** | 81.70 | 157.90 | **53.00** | 44.00 | 64.00 |
| **61 - 70** | 14039 | 12.19% | **14.91** | 5.08 | 41.63 | **104.30** | 77.44 | 131.20 | **117.30** | 87.00 | 159.00 | **52.90** | 44.00 | 63.30 |
| **71 - 80** | 6742 | 5.86% | **15.70** | 5.25 | 44.08 | **91.20** | 67.00 | 119.60 | **112.40** | 85.00 | 150.20 | **52.70** | 43.50 | 63.70 |
| **81 - 90** | 2333 | 2.03% | **16.60** | 5.88 | 46.10 | **84.25** | 64.35 | 112.40 | **106.00** | 79.70 | 142.00 | **53.75** | 44.20 | 64.85 |
| **91 - 100** | 302 | 0.26% | **17.44** | 5.93 | 47.79 | **77.66** | 56.60 | 100.60 | **97.40** | 73.00 | 129.90 | **55.30** | 45.00 | 68.00 |
| **Kruskal-Wallis Sig.** | | | < 0.0001 | | | < 0.0001 | | | < 0.0001 | | | < 0.0001 | | |
| **95%CI Inf. Lim.** | | | < 0.0001 | | | < 0.0001 | | | < 0.0001 | | | < 0.0001 | | |
| **95%CI Sup. Lim.** | | | 0.0030 | | | 0.0016 | | | 0.0019 | | | 0.0001 | | |

Data expressed as median and percentiles. The units of measurement for Lp(a), LDL-C, TG, and HDL-C are expressed in mg/dL.Lp(a), lipoprotein(a); LDL-C, LDL-cholesterol; TG, triglycerides; HDL-C, HDL-cholesterol; CI, Confidence Interval; Inf. Lim., Inferior Limit; Sup. Lim., Superior Limit.

cardiovascular disease. This indicates that Lp(a) assessment alone could classify at least 18% of our sample as having a high cardiovascular risk, regardless of other laboratory exams and comorbidities. Lp(a) levels higher than 70 mg/dL could be found in 13,033 subjects, characterizing 11.3% of the sample, and 7,676 individuals (6.6% of the total) had registered Lp(a) levels higher than 90 mg/dL. The mean levels at the 5th, 10th, 25th, 50th, 75th, 90th and 95th percentiles were 2.42; 2.88; 4.52; 12.92; 36.60; 75.11 and 98.29, respectively (all measured in mg/dL).

Median LDL cholesterol, HDL-cholesterol and triglycerides were 107.2 (interquartile interval 106.2–108.9 mg/dL; p varied from < 0.0001 to 0.021), 53.0 (interquartile interval 58.2–46.0 mg/dL; p varied from < 0.0001 to 0.025) and 98.4 mg/dL (interquartile interval 91.0–112.0 mg/dL; p varied from < 0.0001 to 0.025), respectively.

The lipid profile dataset encompassed LDL-cholesterol measurements of 76,763 subjects, equivalent to 66,6% of the sample. Within this group, a subset of 20,514 individuals (26.7%) displayed LDL-C levels equal to or exceeding 130 mg/dL, as outlined in Table 2. A combination of LDL cholesterol ≥ 130 mg/dL and elevated Lp(a) above 70 mg/dL could be found in 2880 patients (3.7% of the sample).

HDL-cholesterol measurements were accessible for 83,765 individuals within the study sample. Among them, 24,311 individuals (29%) exhibited HDL-cholesterol (HDL-C) levels below the thresholds of 50 mg/dL for women and 40 mg/dL for men, as exhibited in Table 2.

Triglycerides measurements were available for a total of 83,943 patients, equivalent to 72,8% of the total. Elevated triglycerides levels exceeding 150 mg/dL were detected in 21.2% of the patients, corresponding to 17,848 individuals, as detailed in Table 2.

Table 2. Distribution of patients in different Lp(a) cutoffs according to lipid fraction levels.

| Lp(a) Cut-Offs | LDL-C Median/ IQR | LDL-C > 130 | LDL-C < 130 | HDL-C Median/ IQR | HDL-C > 50(M) 40(F) | HDL-C < 50(M) 40(F) | TG Median/ IQR | TG > 150 | TG < 150 |
|---|---|---|---|---|---|---|---|---|---|
| **All** | **107.2** | 56280 | 20514 | **53.00** | 59454 | 24311 | **98.40** | 66095 | 17848 |
| | 84.72 - 132.00 | 100% | 100% | 44.00 −64.00 | 100% | 100% | 71.00 −141.00 | 100% | 100% |
| **=<30** | **105.40** | 40277 | 13419 | **53.00** | 40649 | 17879 | **98.00** | 46000 | 12652 |
| | 83.1 −197.97 | 71.6% | 65.4% | 43.0 - 64.0 | 68.4% | 73.5% | 70.0 −140.90 | 69.6% | 70.9% |
| **> 30** | **111.80** | 16003 | 7095 | **54.00** | 18805 | 6432 | **99.80** | 20095 | 5196 |
| | 88.40 - 136.80 | 28.4% | 34.6% | 45.00 - 65.00 | 31.6% | 26.5% | 73.00 - 139.00 | 30.4% | 29.1% |
| **M.W. or χ²** | < 0.0001 | <0.0001 | | <0.0001 | <0.0001 | | <0.0001 | 0.001 | |
| **=<50** | **106.00** | 46951 | 16051 | **53.00** | 48038 | 20539 | **98.00** | 54093 | 14636 |
| | 83.80 - 130.54 | 83.4% | 78.2% | 43.50 −64.00 | 80.8% | 84.5% | 70.00 - 141.00 | 81.8% | 82.0% |
| **> 50** | **113.02** | 9329 | 4463 | **54.00** | 11416 | 3772 | **101.00** | 12002 | 3212 |
| | 89.00 - 138.89 | 16.6% | 21.8% | 45.20 - 65.00 | 19.2% | 15.5% | 74.00 −140.80 | 18.2% | 18.0% |
| **M.W. or χ²** | < 0.0001 | <0.0001 | | <0.0001 | <0.0001 | | <0.0001 | 0.617 | |
| **=<70** | **106.40** | 50716 | 17634 | **53.00** | 52274 | 22089 | **98.00** | 58735 | 15786 |
| | 84.10 - 130.98 | 90.1% | 86.0% | 43.70 - 64.00 | 87.9% | 90.9% | 70.00 - 140.80 | 88.9% | 88.4% |
| **> 70** | **114.10** | 5564 | 2880 | **54.20** | 7180 | 2222 | **103.30** | 7360 | 2062 |
| | 89.90 - 140.80 | 9.9% | 14.0% | 45.60 - 65.00 | 12.1% | 9.1% | 75.50 - 142.30 | 11.1% | 11.6% |
| **M.W. or χ²** | <0.0001 | <0.0001 | | <0.0001 | <0.0001 | | <0.0001 | 0.117 | |
| **=<90** | **106.66** | 53106 | 18800 | **53.00** | 55228 | 23046 | **98.00** | 61832 | 16609 |
| | 84.40 −131.40 | 94.4% | 91.6% | 43.90 - 64.00 | 92.9% | 94.8% | 70.30– 140.80 | 93.6% | 93.1% |
| **> 90** | **115.00** | 3174 | 1714 | **54.40** | 4226 | 1265 | **105.10** | 4263 | 1239 |
| | 90.4 - 142.10 | 5.6% | 8.4% | 46.00 - 65.00 | 7.1% | 5.2% | 77.00 - 144.00 | 6.4% | 6.9% |
| **M.W. or χ²** | < 0.0001 | <0.0001 | | <0.0001 | <0.0001 | | <0.0001 | 0.018 | |

Data expressed as absolute (n), relative (%) numbers, median and percentiles 25–75.

M.W, Man-Whitney; X², Chi-square; Lp(a), lipoprotein(a); IQR, interquartile range; LDL-C, LDL cholesterol; HDL-C, HDL cholesterol; TG, triglycerides.

DM criteria were positive in 8,267 subjects, leading to a prevalence of 7.2%. Both DM and Lp(a) levels higher than 70 mg/dL were present in 1,096 patients (13.3%), as shown in Table 3.

Three criteria for metabolic syndrome were simultaneously present in 9,214 subjects, from a total of 83,715 individuals, meaning a prevalence of 11%. No criterium for metabolic syndrome was found in 50,843 individuals (60.7%), while 15,080 individuals (18%) showed one criterium and 8,578 individuals (10.2%) had two criteria. The distribution of individuals with metabolic syndrome according Lp(a) cutoffs is presented in Table 3.

To estimate the association between Lp(a) levels, lipid fractions and some clinical conditions, we calculated the odds ratios for different Lp(a) thresholds, given the exposures of interest. Various levels of Lp(a) demonstrated no correlation

**Table 3. Distribution of individuals in different Lp(a) cutoffs, according to the presence of diabetes and number of criteria for metabolic syndrome.**

| Lp(a) Cut-Offs | Metabolic Syndrome | No Criterium (0) | One Criterium (1) | Two Criteria (2) | Three Criteria (3) | p value of Chi-Square |
|---|---|---|---|---|---|---|
| =<30 | n | 34833 | 11042 | 5797 | 6823 | < 0.0001 |
| | % | *68.5%* | *73.2%* | *67.6%* | *74.1%* | |
| > 30 | n | 16010 | 4038 | 2781 | 2391 | |
| | % | *31.5%* | *26.8%* | *32.4%* | *25.9%* | |
| =<50 | n | 41205 | 12741 | 6809 | 7783 | < 0.0001 |
| | % | *81.0%* | *84.5%* | *79.4%* | *84.5%* | |
| > 50 | n | 9638 | 2339 | 1769 | 1431 | |
| | % | *19.0%* | *15.5%* | *20.6%* | *15.5%* | |
| =<70 | n | 44854 | 13724 | 7392 | 8349 | < 0.0001 |
| | % | *88.2%* | *91.0%* | *86.2%* | *90.6%* | |
| > 70 | n | 5989 | 1356 | 1186 | 865 | |
| | % | *11.8%* | *9.0%* | *13.8%* | *9.4%* | |
| =<90 | n | 47340 | 14330 | 7858 | 8699 | < 0.0001 |
| | % | *93.1%* | *95.0%* | *91.6%* | *94.4%* | |
| > 90 | n | 3503 | 750 | 720 | 515 | |
| | % | *6.9%* | *5.0%* | *8.4%* | *5.6%* | |
| Metabolic Syndrome | | Median (25% − 75%) | Median (25% − 75%) | Median (25% − 75%) | Median (25% − 75%) | |
| | | 14.42 (5.17 - 38.5) | 11.6 (4.21 - 32.42) | 14.90 (5.0 - 40.15) | 10.75 (3.77 - 31.42) | |
| Kruskal-Wallis p<0.0001 | | *p<0.001 vs 1, 2 e 3* | *p<0.001 vs 0 e 2* | *p<0.001 vs 0,1 e 3* | *p<0.001 vs 0 e 2* | |

| Lp(a) Cut-Offs | Diabetic | Normal (N) | Pre-diabetic (P) | Diabetic (D) | p value of Chi-Square |
|---|---|---|---|---|---|
| =<30 | n | 39166 | 33240 | 5722 | < 0.0001 |
| | % | *71.8%* | *69.1%* | *69.2%* | |
| > 30 | n | 15365 | 14838 | 2545 | |
| | % | *28.2%* | *30.9%* | *30.8%* | |
| =<50 | n | 45390 | 38829 | 6646 | < 0.0001 |
| | % | *83.2%* | *80.8%* | *80.4%* | |
| > 50 | n | 9141 | 9249 | 1621 | |
| | % | *16.8%* | *19.2%* | *19.6%* | |
| =<70 | n | 49093 | 42139 | 7171 | < 0.0001 |
| | % | *90.0%* | *87.6%* | *86.7%* | |
| > 70 | n | 5438 | 5939 | 1096 | |
| | % | *10.0%* | *12.4%* | *13.3%* | |
| =<90 | n | 51508 | 44524 | 7581 | < 0.0001 |
| | % | *94.5%* | *92.6%* | *91.7%* | |
| > 90 | n | 3023 | 3554 | 686 | |
| | % | *5.5%* | *7.4%* | *8.3%* | |
| Diabetes | | Median (25% − 75%) | Median (25% − 75%) | Median (25% − 75%) | |
| | | 12.13 (4.35-34.6) | 13.83 (4.75 - 38.4) | 12.92 (4.25 - 38.6) | |
| Kruskal-Wallis p<0.0001 | | *p<0.001 vs P e A* | *p<0.001 vs N* | *p<0.001 vs N* | |

Lp(a) levels according to different number of criteria for metabolic syndrome or diabetes are also detailed. Data expressed in absolute (n), relative (%) numbers, median and percentiles.

Lp(a), lipoprotein(a).

with HDL cholesterol, triglycerides, DM, and an increased count of criteria for metabolic syndrome (Table 4). Conversely, a more noticeable association was observed with LDL cholesterol (Table 4).

## Discussion

In this study, our aim was to uncover the number of Lp(a) assessments, estimate Lp(a) mass levels and percentiles, and investigate associations between Lp(a), lipid fractions and some clinical conditions within a large sample from the DASA database.

The major accomplishment of this study lies in its effort to estimate the distribution of Lp(a) levels within a substantial patient sample, encompassing diverse geographical regions and spanning a wide range of ages. This study constitutes an analysis of the most extensive dataset of Lp (a) measurements ever compiled in Brazil.

Through our in-depth analysis of the most comprehensive data set of Lp(a) measurements ever compiled in Brazil, we aim to alleviate the prevailing misconception of Lp(a) elevation as a rare genetic disorder which possibly contributes to its underassessment in clinical practice [12]. It's worth clarifying that Lp(a) dyslipidemia is in reality, the most widespread form of monogenic dyslipidemia, affecting nearly 1 in 5 individuals in the United States when Lp(a) levels exceed 50 mg/dL [13].

This research study effectively addresses the gap in knowledge regarding the prevalence of elevated Lp(a) levels in Brazil. Our findings indicate that 18% of the sampled population of 20,859 patients exceeded the 50 mg/dL threshold for Lp(a), underlining the sizeable proportion of patients who could benefit from cardiovascular risk reclassification and therapy intensification. The underutilization of Lp(a) assessments is likely also influenced by factors such as limited access to laboratory testing, lack of clinician awareness, and the current paucity of approved therapies.

Women constituted the majority of the sample, (n = 70 550 individuals, 61.2% of the total). This is a significant number of Lp(a) requests, considering that Lp (a) is still an overlooked component of risk assessment and that cardiovascular disease (CVD) in women is still understudied, under-recognized, under-diagnosed, and under-treated [13]. This could be related to our cross-sectional, nonintervention observational, laboratory-based study design. Women are more prone to search for diagnostic tests. Despite that, women are notably underrepresented in clinical trials [14] despite CVD being a leading cause of mortality, responsible for 35% of female deaths annually [15].

Women also showed higher Lp(a) levels compared, to men (13.90 vs 11.58 mg/dL, p < 0.001; confidence interval of p < 000.1–0.003). It has been demonstrated that Lp(a) exhibits 5–10% higher levels in women when compared to men, irrespective of race or ethnicity [16–18]. The onset of menopause can contribute to the elevated levels of Lp(a) observed in older women [19]. The specific contribution of these differences in Lp(a) levels to CV outcomes in women is still unclear.

According to the latest EAS and ACC guidelines [3,4], it is recommended to use Lp(a) as a tool to refine cardiovascular risk, especially when levels exceed 50 mg/dL. Additionally, Lp(a) can help identify individuals with an extremely high risk, similar to Heterozygous Familial Hypercholesterolemia, when levels go beyond 180 mg/dL. Based on our dataset, Lp(a) level assessments could potentially alter cardiovascular risk classification for 18% of individuals. Worthy of note, relatives of patients with elevated Lp (a) would benefit from cascade screening strategies (Class IIb recommendation) [4]. Lp(a) is inherited in an autosomal dominant manner, meaning there is a 50% likelihood of identifying a first-degree relative with the condition through cascade screening [20]. This means that a significant number of individuals could benefit of Lp(a) screening, as it would lead to the early initiation of lifestyle changes and treatment intensification [3,4].

Two subsets of patients in our study are worthy of note. First, there were 1,096 diabetic patients with Lp(a) levels above 70 mg/dL. Moreover, a combination of LDL cholesterol < 130 mg/dL and elevated Lp(a) above 70 mg/dL could be found in 2,880 patients. Our hypothesis is that Lp(a) requests for these subpopulations could be relate to primary prevention or secondary prevention strategies in young individuals (possibly some of them under the use of statins).

A growing body of evidence supports a residual risk conferred by Lp(a) in patients within LDL-c goal. In the JUPITER trial, elevated Lp(a) levels were consistently linked to the presence of residual cardiovascular disease risk, even among

**Table 4. Odds ratios for the presence of diabetes and alterations in lipid fractions according to various Lp(a) cutoffs in different age groups.**

| Age Group | Lp(a) cutoff | Diabetes | | | HDL-C < 40 | | | LDL-C ≥ 130 | | | TG > 150 | | |
|---|---|---|---|---|---|---|---|---|---|---|---|---|---|
| | | Odds ratio | 95% Inf. CI | 95% Sup. CI | Odds ratio | 95% Inf. CI | 95% Sup. CI | Odds ratio | 95% Inf. CI | 95% Sup. CI | Odds ratio | 95% Inf. CI | 95% Sup. CI |
| 11 - 20 | 30 | 2.04 | 0.88 | 4.73 | 0.70 | 0.57 | 0.84 | 1.60 | 1.20 | 2.15 | 0.85 | 0.63 | 1.15 |
| | 50 | 0.51 | 0.29 | 0.90 | 0.70 | 0.56 | 0.88 | 1.75 | 1.26 | 2.42 | 0.76 | 0.53 | 1.11 |
| | 70 | 2.40 | 0.88 | 6.55 | 0.67 | 0.50 | 0.90 | 2.15 | 1.48 | 3.13 | 0.77 | 0.48 | 1.22 |
| | 90 | 1.57 | 0.36 | 6.76 | 0.60 | 0.41 | 0.90 | 2.39 | 1.49 | 3.82 | 1.06 | 0.61 | 1.83 |
| 01 - 10 | 30 | 0.66 | 0.62 | 0.70 | 1.04 | 0.64 | 1.69 | 2.10 | 1.03 | 4.25 | 0.99 | 0.46 | 2.12 |
| | 50 | 0.80 | 0.77 | 0.83 | 0.75 | 0.41 | 1.36 | 2.75 | 1.32 | 5.74 | 1.27 | 0.55 | 2.95 |
| | 70 | 0.87 | 0.84 | 0.89 | 0.83 | 0.42 | 1.66 | 3.83 | 1.75 | 8.38 | 0.86 | 0.29 | 2.57 |
| | 90 | 0.93 | 0.90 | 0.95 | 1.10 | 0.47 | 2.55 | 3.81 | 1.46 | 9.90 | 1.25 | 0.36 | 4.35 |
| 21 - 30 | 30 | 1.02 | 0.67 | 1.57 | 0.81 | 0.74 | 0.90 | 1.37 | 1.23 | 1.53 | 0.91 | 0.79 | 1.04 |
| | 50 | 0.93 | 0.60 | 1.45 | 0.80 | 0.70 | 0.90 | 1.53 | 1.34 | 1.75 | 1.05 | 0.89 | 1.24 |
| | 70 | 1.27 | 0.68 | 2.38 | 0.75 | 0.64 | 0.89 | 1.75 | 1.48 | 2.08 | 1.13 | 0.92 | 1.39 |
| | 90 | 0.84 | 0.31 | 2.28 | 0.72 | 0.58 | 0.91 | 1.89 | 1.51 | 2.35 | 1.21 | 0.92 | 1.58 |
| 31 - 40 | 30 | 0.91 | 0.73 | 1.13 | 0.82 | 0.76 | 0.87 | 1.46 | 1.37 | 1.57 | 0.95 | 0.88 | 1.03 |
| | 50 | 1.10 | 0.87 | 1.40 | 0.85 | 0.78 | 0.92 | 1.58 | 1.45 | 1.71 | 1.03 | 0.94 | 1.14 |
| | 70 | 1.07 | 0.78 | 1.48 | 0.80 | 0.72 | 0.89 | 1.82 | 1.64 | 2.01 | 1.08 | 0.96 | 1.22 |
| | 90 | 1.40 | 0.95 | 2.04 | 0.84 | 0.73 | 0.97 | 2.00 | 1.75 | 2.28 | 1.17 | 1.00 | 1.37 |
| 41 - 50 | 30 | 0.90 | 0.78 | 1.04 | 0.77 | 0.71 | 0.82 | 1.29 | 1.20 | 1.39 | 0.87 | 0.80 | 0.94 |
| | 50 | 1.05 | 0.91 | 1.21 | 0.76 | 0.70 | 0.83 | 1.44 | 1.32 | 1.57 | 0.91 | 0.82 | 1.00 |
| | 70 | 0.94 | 0.76 | 1.16 | 0.70 | 0.62 | 0.79 | 1.43 | 1.29 | 1.59 | 0.91 | 0.81 | 1.03 |
| | 90 | 1.00 | 0.77 | 1.30 | 0.63 | 0.54 | 0.73 | 1.52 | 1.32 | 1.74 | 0.89 | 0.76 | 1.04 |
| 51 - 60 | 30 | 0.92 | 0.83 | 1.01 | 0.72 | 0.67 | 0.78 | 1.32 | 1.22 | 1.42 | 0.90 | 0.83 | 0.97 |
| | 50 | 1.10 | 1.00 | 1.21 | 0.67 | 0.61 | 0.74 | 1.38 | 1.27 | 1.50 | 0.93 | 0.85 | 1.02 |
| | 70 | 0.96 | 0.84 | 1.10 | 0.64 | 0.57 | 0.72 | 1.49 | 1.34 | 1.65 | 0.97 | 0.87 | 1.08 |
| | 90 | 1.01 | 0.85 | 1.20 | 0.66 | 0.57 | 0.76 | 1.44 | 1.27 | 1.63 | 1.05 | 0.92 | 1.20 |
| 61 - 70 | 30 | 0.94 | 0.85 | 1.03 | 0.77 | 0.70 | 0.84 | 1.26 | 1.15 | 1.39 | 0.99 | 0.90 | 1.08 |
| | 50 | 1.06 | 0.97 | 1.16 | 0.75 | 0.67 | 0.83 | 1.13 | 1.01 | 1.27 | 0.94 | 0.84 | 1.04 |
| | 70 | 0.93 | 0.82 | 1.05 | 0.72 | 0.63 | 0.81 | 1.18 | 1.03 | 1.34 | 0.97 | 0.86 | 1.10 |
| | 90 | 0.97 | 0.83 | 1.13 | 0.63 | 0.53 | 0.74 | 1.23 | 1.05 | 1.44 | 0.92 | 0.79 | 1.07 |
| 71 - 80 | 30 | 0.92 | 0.82 | 1.04 | 0.87 | 0.76 | 0.99 | 1.16 | 0.99 | 1.35 | 0.81 | 0.70 | 0.93 |
| | 50 | 1.05 | 0.94 | 1.17 | 0.93 | 0.81 | 1.08 | 1.23 | 1.03 | 1.46 | 0.83 | 0.71 | 0.97 |
| | 70 | 0.97 | 0.83 | 1.13 | 0.95 | 0.80 | 1.13 | 1.18 | 0.96 | 1.45 | 0.84 | 0.70 | 1.00 |
| | 90 | 0.96 | 0.79 | 1.17 | 1.03 | 0.84 | 1.26 | 1.00 | 0.77 | 1.29 | 0.78 | 0.62 | 0.97 |
| 81 - 90 | 30 | 1.09 | 0.89 | 1.33 | 0.72 | 0.57 | 0.92 | 1.12 | 0.83 | 1.52 | 0.70 | 0.54 | 0.90 |
| | 50 | 0.93 | 0.79 | 1.10 | 0.77 | 0.59 | 1.01 | 1.14 | 0.81 | 1.60 | 0.75 | 0.55 | 1.00 |
| | 70 | 1.13 | 0.87 | 1.46 | 0.68 | 0.49 | 0.94 | 1.10 | 0.74 | 1.64 | 0.93 | 0.67 | 1.30 |
| | 90 | 1.14 | 0.82 | 1.56 | 0.63 | 0.42 | 0.96 | 1.10 | 0.67 | 1.81 | 1.01 | 0.68 | 1.52 |
| 91 - 100 | 30 | 0.83 | 0.48 | 1.44 | 0.51 | 0.22 | 1.19 | 0.82 | 0.25 | 2.76 | 0.82 | 0.34 | 1.99 |
| | 50 | 1.05 | 0.66 | 1.69 | 0.56 | 0.20 | 1.55 | 0.30 | 0.04 | 2.40 | 0.62 | 0.20 | 1.93 |
| | 70 | 1.09 | 0.55 | 2.16 | 0.44 | 0.13 | 1.56 | 0.46 | 0.06 | 3.68 | 0.96 | 0.30 | 3.01 |
| | 90 | 0.62 | 0.24 | 1.56 | 0.61 | 0.13 | 2.82 | 0.93 | 0.89 | 0.97 | 1.50 | 0.39 | 5.71 |

Lp(a), lipoprotein(a); HDL-C, HDL cholesterol; LDL-C, LDL cholesterol; TG, triglycerides; 95% Inf. C. I., 95% Superior Confidence Interval; 95% Sup. C. I., 95% Superior Confidence Interval.

individuals with low LDL-C levels, with a median of 54 mg/dL [21]. This resonates with an investigation conducted within the Copenhagen General Population, spanning a median follow-up period of 7.4 years, that revealed a heightened risk of major adverse cardiovascular events (MACE) among individuals with Lp(a) levels exceeding 50 mg/dL, even when their LDL-C levels were maintained below 70 mg/dL [22]. These findings indicate that individuals with elevated Lp(a) levels face an elevated risk of recurrent cardiovascular events, even when receiving lipid-lowering therapy. These patients could benefit from targeted therapies designed to lower Lp(a) levels. Comparable results are evident in the latest analysis of the UK Biobank Cohort, focused on a secondary prevention population [6]. This study revealed that a 100 nmol/L elevation in Lp(a) levels correlated with an 8.0% increase in the risk of composite Major Adverse Cardiovascular Events (MACE) and an 18.6% increase in the risk of coronary revascularization throughout the entire follow-up period.

Our investigation revealed a consistent association between elevated Lp(a) levels and LDL-C levels exceeding 130 mg/dL, as outlined in Table 4. These findings harmonize with previous reports [23] indicating that standard LDL-C assays (including methods such as Friedewald or Martin-Hopkins formulas, ultracentrifugation, or direct LDL-C measurements) cannot differentiate between low-density lipoproteins and Lp(a) [18]. Consequently, the reported cholesterol levels effectively encompass the combined cholesterol content of both low-density lipoprotein and Lp (a). Patients with elevated Lp(a) levels will, therefore, exhibit correspondingly elevated LDL-C values when assessed using standard assays. However, it is essential to recognize that these values encompass contributions from both LDL-C and Lp(a), highlighting the intricacies in accurately measuring and interpreting cholesterol profiles. The decision to include LDL- c and HDL-c data from various formulas and techniques in our study is based on two important premises. Firstly, we aim to conduct the study in a real-world setting, with data obtained from different laboratories and techniques commonly used in practice. Secondly, existing literature supports the need to compare different methods of LDL-c estimation, as the Friedewald formula has been shown to underestimate LDL-c levels in certain scenarios [24,25]. By incorporating diverse measurement methods and addressing potential biases, our study will provide a comprehensive analysis that enhances the understanding of LDL-c and HDL-c estimation in real-world settings.

## Clinical implications

The findings of this paper regarding Lp(a) in the Brazilian population have important clinical implications. Lp(a) is known to be a significant risk factor for ASCVD and calcific aortic valve stenosis. Currently, there are challenges in the screening and management of Lp(a) in both primary and secondary prevention. However, this study highlights that in Brazil, the majority of Lp(a) testing is being done for primary prevention. Interestingly, only 18% of the tested population presented with high Lp(a) levels, above 50 mg/dL, which is a lower percentage compared to what is seen in the literature for other countries. For this specific group of patients, it is crucial to consider more aggressive monitoring of cardiovascular risk and intensification of the management of overall risk factors. The results of this study may contribute to future clinical practice recommendations and emphasize the importance of proactive management in individuals with high Lp(a) levels in the Brazilian population.

## Limitations

Our study has some limitations. First, we do not know the motivation driving Lp (a) requests in our sample. Our database lacks information about the health professionals who ordered the exams. Thus, the rationale behind Lp(a) requests could be related to either primary or secondary prevention scenarios, or even nutritional routine assessments. Due to the lack of clinical data, including information about use of statins and other medications, we could not assess cardiovascular risk scores of these patients. Furthermore, this is the main reason we could not estimate the prevalence of metabolic syndrome and relied exclusively on laboratory-base criteria of metabolic syndrome. Lastly, a limitation of this study is the lack of data on ethnicity, which could impact the distribution of Lp(a) in the sample. While there is limited information available on Lp(a) in the Brazilian population, few data from international registries and clinical trial sub analysis, showed that mean

Lp(a) levels tend to be higher in South America, including Brazil, due to the diversity of ethnicity. Non-white patients often have more elevated Lp(a) levels. However, [26,27] it is worth noting that Lp(a) levels were found to be higher in countries with a larger population of black patients, such as the USA, South Africa, and Brazil. Additionally, a Brazilian study investigating the prevalence of the rs10455872 Lp(a) polymorphism as a predictor for prevalent cardiovascular disease showed that ethnicity in Brazil is highly heterogeneous, comprising various ethnic groups, including individuals of European descent, African descent, and Amerindians. While stratified analysis by race did not support a higher prevalence of rs10455872 polymorphism in a specific of ethnic group [28]. In order to address the limitations highlighted in our study, there are several future research avenues that can be explored. In terms of Lp(a) request according to patient profile, collaborations with medical societies and medical education programs could help address these gaps and establish guidelines for the appropriate request and management of Lp(a) testing. Ongoing research in a Brazilian tertiary cardiovascular specialized hospital could also provide valuable insights into Lp(a) in different cardiovascular risk patients and the current treatments for LDL-C.

In conclusion, this is the largest study assessing Lp(a) levels in Brazil. Our findings point to a significant prevalence of Lp(a) elevation, within a large fraction of individuals potentially eligible for risk reclassification and treatment intensification strategies. Moreover, Lp (a) is still overlooked in CV risk assessment, despite being a common genetic disorder. This study fills an important gap in Brazilian literature and adds a relevant contribution to the promotion of awareness among patients and healthcare professionals about the importance of Lp (a) in cardiovascular risk assessment.

## Supporting information

**S1 File. Database.** LP(a) data analysis spreadsheet.
(XLSX)

## Acknowledgments

We express our deepest gratitude to Dr. Felipe Perroni for his invaluable involvement in the early stages of this study. His expertise and guidance were instrumental in shaping the direction and quality of our research.

## Author contributions

**Conceptualization:** Priscila Raupp-da-Rosa, Maria Helane Costa Gurgel Castelo, Patrícia Cristina Grenzi.

**Data curation:** Isac de Castro.

**Formal analysis:** Isac de Castro.

**Investigation:** Maria Helane Costa Gurgel Castelo.

**Methodology:** Priscila Raupp-da-Rosa, Maria Helane Costa Gurgel Castelo, Érica Ferreira.

**Project administration:** Érica Ferreira.

**Resources:** Andreza Almeida Senerchia, Flavia Paiva Proenca Lobo Lopes.

**Software:** Isac de Castro.

**Supervision:** Priscila Raupp-da-Rosa, Eduardo Gomes Lima.

**Validation:** Andreza Almeida Senerchia, Flavia Paiva Proenca Lobo Lopes.

**Visualization:** Andreza Almeida Senerchia, Flavia Paiva Proenca Lobo Lopes.

**Writing – original draft:** Priscila Raupp-da-Rosa, Maria Helane Costa Gurgel Castelo, Isac de Castro, Eduardo Gomes Lima, Flavia Paiva Proenca Lobo Lopes.

**Writing – review & editing:** Priscila Raupp-da-Rosa, Maria Helane Costa Gurgel Castelo, Eduardo Gomes Lima.

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
