## [Decision Letter · Decision Letter 0]

Dear Dr. Raupp-da-Rosa,

Thank you for submitting your manuscript to PLOS ONE. After careful consideration, we feel that it has merit but does not fully meet PLOS ONE’s publication criteria as it currently stands. Therefore, we invite you to submit a revised version of the manuscript that addresses the points raised during the review process.

We look forward to receiving your revised manuscript.

Kind regards,

Hean Teik Ong

Academic Editor

PLOS ONE

**Journal Requirements:**

Novartis Biociências SA provided financial support for the conduct of the research, preparation of the article, study design, critical analysis, interpretation of data, writing of the report, and the decision to submit the article for publication. PR, MHC, IC, PCG, EGL, AAS, EF, and FPPLL received financial support from Novartis.

I have read the journal's policy and the authors of this manuscript have the following competing interests: PR and EF are employees of Novartis Biociências SA. EGL reports that Novartis has made payments to their current employer. FPPLL has received direct payment from Bayer.

MHCGC, IC, PCG, and AAS have declared that no competing interests exist.

We note that one or more of the authors are employed by a commercial company.

“The funder provided support in the form of salaries for authors, but did not have any additional role in the study design, data collection and analysis, decision to publish, or preparation of the manuscript. The specific roles of these authors are articulated in the ‘author contributions’ section.”

**Additional Editor Comments:**

Please make major revisions to address comments of the reviewers and administrative editor.

Administrative editor:

1.Please ensure that this manuscript meets PLOS ONE's guidelines on observational studies involving human subjects https://journals.plos.org/plosone/s/submission-guidelines.

2.The authors have declared the following potential competing interest: PR and EF are employees of Novartis Biociências SA. EGL reports that Novartis has made payme. Please take extra care to ensure that the data and conclusions are presented objectively and in an unbiased manner.

**Reviewer 1:**

Data from this study is relevant to the exploration of Lp(a), particularly prevalence and differences in Lp(a) levels in different populations, and correlation with MetS components which is still not exhaustively reported. However, the following are queries to be clarified by the author to further enhance the study findings:

1. The study design needs to be standardized. In the Introduction section, it is mentioned that the study design is a retrospective observational study. However, in methodology, the author states that it is cross-sectional, non-intervention observational study.

2. What was the Metabolic Syndrome criteria did the author adopt to identify subjects with MetS? This was not described in the methodology which is worthwhile incorporating to give more clarity to the method.

3. Based on the description of determining LDL-c, there are 3 methods which is direct measurement (utilizing 6 equipment), LDL-c calculations by Friedewald and Martin-Hopkins formulas. How do you ensure correlation of results between the 6 equipment? How do you determine the agreement between the direct measurement and calculated LDL-c? Are the 2 equations well correlated?

4. Similarly with HDL-c measurement between the equipment. How do you determine the instruments correlated with one another?

5. How did the author minimized any bias due to ethnic differences in the Brazilian population, considering that Brazil has approximately 5-7 major ethnic groups from literatures? Worthwhile exploring this or highlighting how any potential bias was overcome.

6. Discussion section:

a) The sentences in lines 271-275 are unclear in terms of what the message the author aims to deliver. Please clarify.

b) Sentence in lines 286-287: 18% of study population refers to what lipid patterns?

c) What are the future research that can be done to close the gaps of the limitations you've highlighted in the discussion?

Minor comments:

1. To include quality controls and whether method verification was done prior to the analyses of the biomarkers since the results are heavily lab based and therefore ensuring the reliability of data is important.

2. Grammatical errors which is recommended to have the manuscript undergo grammar check, particularly in the discussion section.

3. Lp(a) and lipoprotein (a) terms are used interchangeably. Suggest to use Lp(a) throughout once the abbreviation is defined in the introduction.

**Reviewer 2:**

This is a good study using real-world data. But I think it is important to highlight/acknowledge the limitations. There is a lack of clinical data (medication use, whether treated for diabetes or hyperlipidemia/statin and specific reasons for Lp(a) testing. This limits the ability to fully assess cardiovascular risk and patient management.

There is also no mention of ethnicity in the study.

I would recommend to

1.Expand discussion on limitations

2.Compare with different ethnicity among Brazilians

3.Any potential clinical implications

4.Add statement on future research direction

5.Language is a bit lengthy. Perhaps can be more direct and concise. Some parts are repetitive (example line 28-31, 32-35, 67-71, spelling error line 96. Confidence interval can be abbreviated to CI.

6.Table 2 is difficult to read and understand. Table 3 Suggest adjust column width so that percentages can be one line.

Reviewers' comments:

Reviewer's Responses to Questions

**Comments to the Author**

1. Is the manuscript technically sound, and do the data support the conclusions?

Reviewer #1: Partly

Reviewer #2: Yes

2. Has the statistical analysis been performed appropriately and rigorously?

Reviewer #1: Yes

Reviewer #2: Yes

3. Have the authors made all data underlying the findings in their manuscript fully available?

Reviewer #1: Yes

Reviewer #2: Yes

4. Is the manuscript presented in an intelligible fashion and written in standard English?

Reviewer #1: Yes

Reviewer #2: No

**Reviewer #1: ** Data from this study is relevant to the exploration of Lp(a), particularly prevalence and differences in Lp(a) levels in different populations, and correlation with MetS components which is still not exhaustively reported. However, the following are queries to be clarified by the author to further enhance the study findings:

1. The study design needs to be standardized. In the Introduction section, it is mentioned that the study design is a retrospective observational study. However, in methodology, the author states that it is cross-sectional, non-intervention observational study.

2. What was the Metabolic Syndrome criteria did the author adopt to identify subjects with MetS? This was not described in the methodology which is worthwhile incorporating to give more clarity to the method.

3. Based on the description of determining LDL-c, there are 3 methods which is direct measurement (utilizing 6 equipment), LDL-c calculations by Friedewald and Martin-Hopkins formulas. How do you ensure correlation of results between the 6 equipment? How do you determine the agreement between the direct measurement and calculated LDL-c? Are the 2 equations well correlated?

4. Similarly with HDL-c measurement between the equipment. How do you determine the instruments correlated with one another?

5. How did the author minimized any bias due to ethnic differences in the Brazilian population, considering that Brazil has approximately 5-7 major ethnic groups from literatures? Worthwhile exploring this or highlighting how any potential bias was overcome.

6. Discussion section:

a) The sentences in lines 271-275 are unclear in terms of what the message the author aims to deliver. Please clarify.

b) Sentence in lines 286-287: 18% of study population refers to what lipid patterns?

c) What are the future research that can be done to close the gaps of the limitations you've highlighted in the discussion?

Minor comments:

1. To include quality controls and whether method verification was done prior to the analyses of the biomarkers since the results are heavily lab based and therefore ensuring the reliability of data is important.

2. Grammatical errors which is recommended to have the manuscript undergo grammar check, particularly in the discussion section.

3. Lp(a) and lipoprotein (a) terms are used interchangeably. Suggest to use Lp(a) throughout once the abbreviation is defined in the introduction.

**Reviewer #2:**  This is a good study using real-world data. But I think it is important to highlight/acknowledge the limitations. There is a lack of clinical data (medication use, whether treated for diabetes or hyperlipidemia/statin and specific reasons for Lp(a) testing. This limits the ability to fully assess cardiovascular risk and patient management.

There is also no mention of ethnicity in the study.

I would recommend to

1.Expand discussion on limitations

2.Compare with different ethnicity among Brazilians

3.Any potential clinical implications

4.Add statement on future research direction

5.Language is a bit lengthy. Perhaps can be more direct and concise. Some parts are repetitive (example line 28-31, 32-35, 67-71, spelling error line 96. Confidence interval can be abbreviated to CI.

6.Table 2 is difficult to read and understand. Table 3 Suggest adjust column width so that percentages can be one line.

**Do you want your identity to be public for this peer review?** For information about this choice, including consent withdrawal, please see our Privacy Policy

Reviewer #1: No

Reviewer #2: No

---

## [Author Response · Author response to Decision Letter 1]

28 Feb 2025

Dear reviewers,

Thank you for dedicating your time and expertise to provide valuable feedback on our manuscript. We greatly appreciate your insightful suggestions, which have helped us improve the quality of our work. Below, we have provided detailed responses to each of your comments and indicated where revisions were made in the manuscript. Your feedback has been invaluable, and we have strived to address all points comprehensively.

Journal Requirements:

1. Please provide an amended statement that declares *all* the funding or sources of support (whether external or internal to your organization) received during this study. Please also include the statement “There was no additional external funding received for this study.” in your updated Funding Statement. Please also include your amended Funding Statement within your cover letter.

Thank you for reminding us about the necessary details in the funding description. We have provided the requested detailed information in our cover letter, as well as in the submission system as follows: “Novartis Biociências SA provided financial support for the conduct of the research, preparation of the article, study design, critical analysis, interpretation of data, writing of the report, and the decision to submit the article for publication. There was no additional external funding received for this study.”

2. Please provide an amended Funding Statement and Competing Interests Statement, including details about the authors' commercial affiliations, their roles in the study, and any restrictions on sharing data and materials. Confirm that the commercial affiliation does not affect adherence to PLOS ONE’s policies on data sharing. “

We have carefully reviewed and updated these sections in accordance with your guidance as follow: “The authors have disclosed the following potential conflicts of interest: PR is an employee and stockholder of Novartis Biociências SA. PR also holds a leadership role as an Emergent Leader in the World Heart Federation. EGL reports that Novartis made payments to DASA, their current employer, in relation to this manuscript. EGL also received honoraria as a speaker for Novartis in educational events and has served on an advisory board for Bayer. FPPLL received direct payments from Bayer for attending meetings or travel. EF is a salaried employee of Novartis Biociências SA. IC received consulting fees through their institution, ConstatBio, related to DASA. MHCGC, PCG, and AAS declared no competing interests. These commercial affiliations do not alter our adherence to PLOS ONE's policies on data and material sharing.”

Reviewers’ comments

Reviewer 1:

1. The study design needs to be standardized. In the Introduction section, it is mentioned that the study design is a retrospective observational study. However, in methodology, the author states that it is cross-sectional, nonintervention observational study.

Thank you for highlighting this discrepancy in the study design description. We have corrected the manuscript to consistently describe the study design as a retrospective observational study in both the Introduction (page 1, paragraph 1) and Methods sections (page 1, paragraph 2).

2. What was the Metabolic Syndrome criteria did the author adopt to identify subjects with MetS? This was not described in the methodology which is worthwhile incorporating to give more clarity to the method.

Thank you for your comment regarding the clarification of the MetS criteria used in our study. The criteria is described on page 4, lines 77-80: “Based on US National Cholesterol Education Program (NCEP) definitions [10], the following metabolic syndrome laboratorial criteria were considered: triglycerides > 150 mg/dL; fasting plasma glucose > 110 mg/dL; HDL-cholesterol < 40 mg/dL (men), HDL- cholesterol < 50 mg/dL (women).”

At page 20 and 21 lines 341- 344, we clarify in the discussion that we used only laboratory criteria to Metabolic Syndrome definition -. “Furthermore, this is the main reason we could not estimate the prevalence of metabolic syndrome and relied exclusively on laboratory-base criteria of metabolic syndrome. Lastly, a limitation of this study is the lack of data on ethnicity, which could impact the distribution of Lp(a) in the sample.”

3. Based on the description of determining LDL-c, there are 3 methods which is direct measurement (utilizing 6 equipment), LDL-c calculations by Friedewald and Martin-Hopkins formulas. How do you ensure correlation of results between the 6 equipment? How do you determine the agreement between the direct measurement and calculated LDLc? Are the 2 equations well correlated?

Thank you for your valuable questions regarding the methods used to determine LDL-c levels and for your thoughtful inquiry into the correlation and agreement between the different techniques and formulas. The decision to include LDL data measured by different formulas and techniques in our study stems from two important premises. Premise 1 highlights that this study was conducted in a real-world setting, utilizing the database of the largest clinical laboratory in Latin America. Unlike a randomized clinical trial, our objective was to capture data from different laboratories and techniques, reflecting the diversity of test methods commonly used in practice.

Premise 2 is based on existing literature, which presents publications from different countries comparing LDL levels calculated by modified Martin/Hopkins estimation or the Friedewald formula with direct homogeneous assay measured low-density lipoprotein cholesterol. It has been observed that the Friedewald equation tends to underestimate LDL-C levels in very high and high-risk settings. An analysis conducted on Hungarian patients concluded that LDL-C estimation using the Martin/Hopkins formula, which is validated by the beta-quantification method, yields a more accurate LDL-C value than that calculated by the Friedewald formula.

By incorporating the data obtained through various measurement methods, particularly with consideration of the limitations of the Friedewald formula, our study aims to provide a comprehensive analysis that reflects real-world scenarios and contributes to the understanding of LDL-C estimation. By doing so, we can gain insights into the accuracy and reliability of different measurement techniques and highlight the importance of using validated formulas in specific clinical contexts.

It is important to acknowledge that the inclusion of data from different formulas and techniques may introduce variability. However, through appropriate statistical analyses and quality control measures, we can account for these factors and provide valuable information on the correlation, agreement, and potential biases associated with different measurement methods. Ultimately, this approach strengthens the validity and applicability of our study findings, offering a more robust understanding of LDL-C estimation in real-world settings. A paragraph was added to the article in the discussion section. It is presented on page 19, lines 311–319: "The decision to include LDL- c and HDL-c data from various formulas and techniques in our study is based on two important premises. Firstly, we aim to conduct the study in a real-world setting, with data obtained from different laboratories and techniques commonly used in practice. Secondly, existing literature supports the need to compare different methods of LDL-c estimation, as the Friedewald formula has been shown to underestimate LDL-c levels in certain scenarios [25,26]. By incorporating diverse measurement methods and addressing potential biases, our study will provide a comprehensive analysis that enhances the understanding of LDL-c and HDL-c estimation in real-world settings."

4. Similarly with HDL-c measurement between the equipment. How do you determine the instruments correlated with one another?

Thank you for your follow-up question regarding the measurement of HDL-c and the correlation between the equipment used. As we explained in our response to Question 3, the approach to ensuring correlation and agreement between the different instruments and measurement techniques applies similarly to HDL-c as it does to LDL-c. The decision to include data from different techniques, based on real-world data from the largest clinical laboratory in Latin America, reflects the diversity of test methods commonly used in clinical practice. This strategy allows for a more comprehensive analysis of the methods' accuracy and reliability. We hope this explanation helps clarify the methodology and thank you again for your attention to detail.

5. How did the author minimized any bias due to ethnic differences in the Brazilian population, considering that Brazil has approximately 5-7 major ethnic groups from literatures? Worthwhile exploring this or highlighting how any potential bias was overcome.

Thank you for your question regarding bias due to ethnic. This aspect has been clarified in the text on page 20 and 21, lines 343-362. Here are the key points that address this concern:

The study utilized data from a national database of a private Brazilian laboratory, which included a wide range of ages and geographic regions. This approach ensured a representative sample of the Brazilian population, known for its ethnic diversity. The geographic analysis of patient locations revealed that individuals were mainly concentrated in urban centers such as São Paulo, Rio de Janeiro, Brasília, Curitiba, Recife, and Fortaleza. This suggests that the sample included individuals from diverse ethnic backgrounds, reflecting Brazil's demographic composition.

Although the study did not collect specific data on ethnicity, it acknowledges Brazil's ethnic heterogeneity and the possible influence of this diversity on Lp(a) levels. International studies indicate that Lp(a) levels tend to be higher in non-white populations, which includes a significant portion of the Brazilian population. The document references previous studies that investigated the prevalence of genetic polymorphisms associated with Lp(a) in different ethnic groups in Brazil. These studies showed that ethnicity in Brazil is highly heterogeneous, including individuals of European, African, and Amerindian ancestry.

The study also acknowledges its limitations, including the lack of data on ethnicity, and suggests that future research should explore the influence of ethnic diversity on Lp(a) levels in the Brazilian population in more detail. This includes collaborations with medical societies and medical education programs to establish appropriate guidelines for requesting and managing Lp(a) tests.

These points demonstrate that the author took measures to minimize bias due to ethnic differences by ensuring a representative sample and recognizing the need for future research to address this issue in more detail.

6. The sentences in lines 271-275 are unclear in terms of what the message the author aims to deliver. Please clarify.

We appreciate your feedback and have revised the text to ensure the message is clear and concise. The updated version is as follows: “According to the latest EAS and ACC guidelines [3,4], it is recommended to use Lp(a) as a tool to refine cardiovascular risk, especially when levels exceed 50 mg/dL. Additionally, Lp(a) can help identify individuals with an extremely high risk, similar to Heterozygous Familial Hypercholesterolemia, when levels go beyond 180 mg/dL.” (Page 17, lines 467 – 470).

7. Sentence in lines 286-287: 18% of study population refers to what lipid patterns?

Thank you for your observation. We clarify the information on page 10, lines 180 – 184: “Based on our dataset, we found that 18% of the individuals (20,859 subjects) had Lp(a) levels above 50 mg/dL, which is considered a high risk for cardiovascular disease. This indicates that Lp(a) assessment alone could classify at least 18% of our sample as having a high cardiovascular risk, regardless of other laboratory exams and comorbidities.”

8. What are the future research that can be done to close the gaps of the limitations you've highlighted in the discussion?

Thank you for your thoughtful question. We have added future research recommendations on page 21, lines 355 - 362: “In order to address the limitations highlighted in our study, there are several future research avenues that can be explored. In terms of Lp(a) request according to patient profile, collaborations with medical societies and medical education programs could help address these gaps and establish guidelines for the appropriate request and management of Lp(a) testing. Ongoing research in a Brazilian tertiary cardiovascular specialized hospital could also provide valuable insights into Lp(a) in different cardiovascular risk patients and the current treatments for LDL-C.”

9. To include quality controls and whether method verification was done prior to the analyses of the biomarkers since the results are heavily lab based and therefore ensuring the reliability of data is important.

Thank you for your comment. Here are the relevant points that address this concern:

The study utilized data from a national database of a private Brazilian laboratory, Diagnósticos da América S.A. (DASA), which ensured a comprehensive and standardized approach to laboratory measurements. The following quality controls and method verifications were implemented:

1. Laboratorial Measurements: The biomarkers, including Lp(a), fasting blood glucose, A1c, total plasma cholesterol, HDL cholesterol, LDL cholesterol, and triglycerides, were measured using standardized methods and equipment. For instance, Lp(a) was measured in mg/dL by nephelometric assay using Cobas 8000, Imuno BNII Cluster 1, Cluster 2, and Main Cluster devices

2. Reference Intervals and Methods: Each biomarker was measured using specific methods with established reference intervals. For example, fasting plasma glucose was evaluated by the hexokinase method, and A1c was measured using various devices such as Advia Cluster, Cobas, Premier Hb9210, Tosoh Cluster, Cluster GHLV, Hedmato D-100, and Cobas C513. These methods ensured consistency and accuracy in the measurements.

3. Quality Control Procedures: The study adhered to rigorous quality control procedures to ensure the reliability of the data. This included the use of multiple devices and methods to cross-verify the results. For instance, triglycerides were measured using an enzymatic colorimetric method with cholesterol esterase oxidase, and LDL-C was calculated using both the Friedewald formula and the Martin-Hopkins formula

4. Statistical Analysis: The study employed robust statistical methods to analyze the data, including the use of the Kolmogorov-Smirnov test, Mann-Whitney test with Bonferroni correction, Kruskal-Wallis test, and Monte Carlo correction with 1,000 resampling

5. Ethical Considerations: The study followed ethical principles and obtained a waiver of informed consent due to the retrospective nature of the study. All data was anonymized and encrypted to protect patient privacy.

These points demonstrate that the study included comprehensive quality controls and method verifications to ensure the reliability of the data, given the lab-based nature of the results.

10. Grammatical errors which is recommended to have the manuscript undergo grammar check, particularly in the discussion section.

Thank you for your feedback regarding the language used in the manuscript. We have reviewed and improved the manuscript.

11. Lp(a) and lipoprotein (a) terms are used interchangeably. Suggest to use Lp(a) throughout once the abbreviation is defined in the introduction.

Thank you for your valuable suggestion regarding consistent terminology. We have ensured that the term "Lp(a)" is used consistently throughout the manuscript after its definition in the introduction.

Reviewer 2:

1. Expand discussion on limitations;

Thank you for your suggestion to expand the discussion on limitations. We have addressed this by elaborating on a dedicated section called “Limitations”, where we provide a more in-depth analysis of the study’s constraints and their potential implications.

2. Compare with different ethnicity among Brazilians;

Thank you for your suggestion to include a c

---

## [Decision Letter · Decision Letter 1]

Dear Dr. Raupp-da-Rosa,

Thank you for submitting your manuscript to PLOS ONE. After careful consideration, we feel that it has merit but does not fully meet PLOS ONE’s publication criteria as it currently stands. Therefore, we invite you to submit a revised version of the manuscript that addresses the points raised during the review process.

We look forward to receiving your revised manuscript.

Kind regards,

Chiara Pavanello

Academic Editor

PLOS ONE

Journal Requirements:

Reviewers' comments:

Reviewer's Responses to Questions

**Comments to the Author**

Reviewer #3: All comments have been addressed

2. Is the manuscript technically sound, and do the data support the conclusions?

Reviewer #3: Yes

3. Has the statistical analysis been performed appropriately and rigorously?

Reviewer #3: Yes

4. Have the authors made all data underlying the findings in their manuscript fully available?

Reviewer #3: Yes

5. Is the manuscript presented in an intelligible fashion and written in standard English?

Reviewer #3: Yes

Reviewer #3: The manuscript has been thoroughly revised, and the authors have addressed the reviewers’ comments. I have only a few minor remarks:

Abstract: please include the p-value for the comparison between men and women.

Introduction, line 59: The phrase “secondary cause of Lp(a) levels alterations” needs clarification. Could the authors specify which secondary causes are being referred to?

Lines 87, 88, 90, 108: The unit mg/dL is missing in several instances. Additionally, A1c should be referred to as HbA1c, and it is not expressed in mg/dL. Please remove mg/dL from line 88 and correct the terminology in line 90 accordingly.

The US NCEP/ATP III cut-off values are inaccurately reported. Triglycerides should be defined as ≥150 mg/dL, and fasting glucose as >110 mg/dL. Please revise these values.

HbA1c: Ensure that "A1c" is consistently reported as HbA1c throughout the manuscript.

In the expression “p varied from...,” if the authors are referring to a confidence interval, this should be reported as: p < 0.0001; 95% CI: 0.0001–0.003.

Tables:

Please use a dot (.) instead of a comma (,) for decimal separation.

Units of measurement are missing in Table 1 and should be added.

There are two tables labeled as Table 2—please correct the numbering.

Please check and revise the legend of Table 3 for clarity and consistency.

Lines 256–258: The statement regarding the “misperception of high Lp(a) as a rare genetic disorder” is unclear and potentially misleading. Why should elevated Lp(a) be perceived as a rare genetic disorder? However, I do not believe this is the main reason for the limited assessment of Lp(a) in clinical practice. Rather, it likely reflects restricted access to laboratory testing, limited awareness among clinicians, and—perhaps most importantly—the current lack of approved therapies.

**Do you want your identity to be public for this peer review?** For information about this choice, including consent withdrawal, please see our Privacy Policy

Reviewer #3: No

---

## [Author Response · Author response to Decision Letter 2]

20 May 2025

Dear Reviewer,

We sincerely thank you for your careful and constructive evaluation of our manuscript. Your insightful comments were extremely helpful in improving the clarity, precision, and overall quality of our work. Below, we provide a detailed point-by-point response, along with a description of the corresponding changes made to the manuscript.

Review Comments to the Author

Reviewer #3

1. Abstract: please include the p-value for the comparison between men and women.

Thank you for your observation regarding the Abstract. As requested, we have added the p-value for the comparison between men and women to improve clarity and completeness of the statistical reporting.

2. Introduction, line 59: The phrase “secondary cause of Lp(a) levels alterations” needs clarification. Could the authors specify which secondary causes are being referred to?

Thank you for pointing out the ambiguity in our phrase “secondary cause of Lp(a) levels alterations.” We appreciate your diligent attention to detail.

The secondary causes we are referring to in this context include certain health conditions and treatments that can affect Lp(a) levels. For instance, physical activity and exercise can influence Lp(a) levels, though the extent varies low values depending on factors such as age, type, intensity, and duration of the exercise.

Hormone replacement therapy (HRT) can reduce Lp(a) levels in postmenopausal women, with oral administration being more effective than transdermal estradiol. The Lp(a)-lowering effect of HRT is not influenced by the type of HRT, the dose of estrogen, or the addition of progestogen.

Kidney diseases can cause increases in Lp(a) levels, depending on disease stages, dialysis methods, and apolipoprotein(a) phenotypes. Conversely, liver diseases may lower Lp(a) levels in tandem with disease progression, although there have been conflicting results about the association between Lp(a) levels and nonalcoholic fatty liver disease in population studies. We modified in the text this sentence.

3. Lines 87, 88, 90, 108: The unit mg/dL is missing in several instances. Additionally, A1c should be referred to as HbA1c, and it is not expressed in mg/dL. Please remove mg/dL from line 88 and correct the terminology in line 90 accordingly.

Thank you for your attentive review and valid suggestions. We have taken your comments into consideration and made the necessary amendments in the text. Instances that were missing the unit mg/dL have been corrected and the inaccurate use of A1c is now appropriately referred to as HbA1c. The term mg/dL has been duly removed from line 88.

4. The US NCEP/ATP III cut-off values are inaccurately reported. Triglycerides should be defined as ≥150 mg/dL, and fasting glucose as >110 mg/dL. Please revise these values.

Thank you for your keen observation regarding the inaccuracies in the reported cut-off values for triglycerides and fasting glucose according to the US NCEP/ATP III. We have amended these values in our text, with triglycerides now correctly defined as ≥150 mg/dL and fasting glucose accurately stated as >110 mg/dL. Your precise attention to detail helps maintain the integrity and accuracy of our study.

5. HbA1c: Ensure that "A1c" is consistently reported as HbA1c throughout the manuscript.

Thank you for your attentive review and valid suggestions. We have been corrected and the inaccurate use of A1c is now appropriately referred to as HbA1c.

6. In the expression “p varied from...,” if the authors are referring to a confidence interval, this should be reported as: p < 0.0001; 95% CI: 0.0001–0.003.

The reviewer's question is very important and extremely pertinent regarding this unusual form of analysis.

One of the constant criticisms made when comparing continuous or categorical data in large samples is that significant differences occur between samples with minimum mean/median values that have no clinical relevance. The same occurs with the comparison between rates that may present significant differences between minimum percentage differences that also have no clinical relevance.

To eliminate this statistical bias, in the comparisons mentioned by the reviewer, an important resource was used: the analysis of random subsamples within the samples studied. In this case, the standard of one thousand subsamples was used, which, because they have a smaller number of individuals, mitigates this bias and also excludes the possibility of potential subsamples with a different behavior from the sample as a whole. Therefore, for each comparison of these subsamples, a significance value (p) is generated, resulting in a thousand significance values. To determine whether any of these sub-analyses have a discrepancy, the confidence intervals of the thousand significances analyzed are calculated. For example: if a lower confidence interval value were found with a significance of less than 0.05 while the upper confidence interval value was greater than 0.05, we would have a null statistical analysis.

An alternate method for determination of the accuracy of the model parameters is the so-called “bootstrap” method (Efron, 1979; Efron and Tibshirani, 1993; Shao and Tu, 1996).

Andrych et al. (2020) and Andronov and Kulynska (2020) discussed statistical properties of the approximations of the “bootstrap-generated” data sets in more detail.

References (bootstrapping algorithms)

• Sheen Mclean Cabaneros, Ben Hughes, Methods used for handling and quantifying model uncertainty of artificial neural network models for air pollution forecasting,Environmental Modelling & Software,Volume 158,2022,105529,ISSN 1364-8152, (https://doi.org/10.1016/j.envsoft.2022.105529)

• Davison, A. C., and D. V. Hinkley. 2006. Bootstrap Methods and their Application. : Cambridge University Press.

• Shao, J., and D. Tu. 1995. The Jackknife and Bootstrap. New York: Springer.

7. Lines 256–258: The statement regarding the “misperception of high Lp(a) as a rare genetic disorder” is unclear and potentially misleading. Why should elevated Lp(a) be perceived as a rare genetic disorder? However, I do not believe this is the main reason for the limited assessment of Lp(a) in clinical practice. Rather, it likely reflects restricted access to laboratory testing, limited awareness among clinicians, and—perhaps most importantly—the current lack of approved therapies.

Thank you for your thought-provoking comments. We appreciate your constructive perspective.

Our objective with the statement regarding the "misperception of high Lp(a) as a rare genetic disorder" was to highlight the common misconception that elevated Lp(a) levels are confined to a small group of individuals with genetic disorders. This overlooks the fact that elevated Lp(a) levels are more widespread, affecting nearly 1 in 5 individuals when levels surpass 50 mg/dL.

That being said, you are absolutely right that the limited assessment of Lp(a) in clinical practice may also be due to factors such as restricted laboratory testing, limited awareness of Lp(a) among clinicians, and the lack of approved therapies specifically targeting Lp(a).

In light of your feedback, we will refine our thesis. We aim to impart that misconceptions about the rarity of Lp(a) dyslipidemia, as well as the factors you outlined, significantly contribute to the underutilization of Lp(a) assessments in routine clinical practice.

8. Tables: Please use a dot (.) instead of a comma (,) for decimal separation.

Thank you for pointing that out. We have standardized the decimal format throughout the tables, replacing commas with dots, as per your suggestion.

9. Units of measurement are missing in Table 1 and should be added.

Thank you for your observation. Units of measurement were added to the table caption.

10. There are two tables labeled as Table 2—please correct the numbering.

Thank you for your comment. We reviewed the table numbers to ensure clarity and consistency.

11. Please check and revise the legend of Table 3 for clarity and consistency.

Thank you for your observation. The table was redesigned to improve clarity and consistency.

---

## [Editor Report · Decision Letter 2]

Lipoprotein(a) levels in a sample of 115,197 subjects from the largest Brazilian private laboratory

PONE-D-24-52143R2

Dear Dr. Raupp-da-Rosa,

We’re pleased to inform you that your manuscript has been judged scientifically suitable for publication and will be formally accepted for publication once it meets all outstanding technical requirements.

Kind regards,

Chiara Pavanello

Academic Editor

PLOS ONE
---

## [Editor Report · Acceptance letter]

PONE-D-24-52143R2

PLOS ONE

Dear Dr. Raupp-da-Rosa,

I'm pleased to inform you that your manuscript has been deemed suitable for publication in PLOS ONE. Congratulations! Your manuscript is now being handed over to our production team.

Kind regards,

on behalf of

Dr. Chiara Pavanello

Academic Editor

PLOS ONE